# On Union-Closedness of Language Generation

**Steve Hanneke**
Purdue University
steve.hanneke@gmail.com

**Anay Mehrotra**
Yale University
anaymehrotra1@gmail.com

**Amin Karbasi**
Yale University
amin.karbasi@gmail.com

**Grigoris Velegkas**[*]
Google Research
gvelegkas@google.com

## Abstract

We investigate language generation in the limit – a model by Kleinberg and Mullainathan [2024, NeurIPS] and extended by Li, Raman, and Tewari [2025]. While Kleinberg and Mullainathan proved generation is possible for all countable collections, [Li et al., 2025] defined a hierarchy of generation notions (uniform, non-uniform, and generatable) and explored their feasibility for uncountable collections. Our first set of results resolve two open questions of [Li et al., 2025] by proving finite unions of generatable or non-uniformly generatable classes need not be generatable. These follow from a stronger result: there is a non-uniformly generatable class and a uniformly generatable class whose union is non-generatable. This adds to the aspects along which language generation in the limit is different from traditional tasks in statistical learning theory like classification, which are closed under finite unions. In particular, it implies that given two generators for different collections, one cannot combine them to obtain a single "more powerful" generator, prohibiting this notion of boosting. Our construction also addresses a third of [Li et al., 2025]'s open questions on whether there are uncountable classes that are non-uniformly generatable and do not satisfy the eventually unbounded closure (EUC) condition introduced by Li, Raman, and Tewari. Our approach utilizes carefully constructed classes along with a novel diagonalization argument that could be of independent interest in the growing area of language generation.

## 1 Introduction

The algorithmic problem at the core of language generation – in both humans and large language models (LLMs) – is deceptively simple: given a sequence of examples from some target language, generate new and previously unseen strings that also belong to this language. Despite the remarkable capabilities of humans and, perhaps, more so of LLMs, to generate coherent text, a thorough understanding of the theoretical underpinnings of language generation has remained elusive.

**Language Generation in the Limit.** Kleinberg and Mullainathan [2024] recently formalized this problem in a model resembling online learning: First, an adversary fixes a target language $K \in \mathcal{L}$ and an enumeration of $K$.[2] At each round $n \geq 1$, the adversary presents the $n$-th element $x_n$ of the enumeration to the generator. The generator, given the strings $S_n = \{x_1, \ldots, x_n\}$ seen so far, outputs a new string $w_n \notin S_n$ – its guess for an unseen string in $K$. The generator wins this game if it eventually learns "to generate from $K$." Formally, a generator $G$ is said to generate from $\mathcal{L}$ in

---

[*]Part of the work was done while the author was a PhD student at Yale University.

[2]Formally, en enumeration of $K$ is an infinite sequence of elements $x_1, x_2, \ldots$ (possibly including duplicates) such that each $x_i \in K$, and for every element $x \in K$ there is some position $n_x$ in the sequence where $x$ appears.

the limit if for all $K \in \mathcal{L}$ and any enumeration of $K$, there exists a finite time $n^\star$ such that for any subsequent round $n \geq n^\star$, the generated string $w_n$ is an unseen element of $K$, *i.e.*, $w_n \in K \setminus S_n$.

This model is closely connected to the seminal work of Gold [1967] – which studied the problem with the harder goal of *identifying* the target language $K$ from a collection of languages $\mathcal{L}$ – and sparked a long line of work both in linguistics and computer science, culminating in a complete characterization by Angluin [1979, 1980]. This characterization revealed a profound paradox: while language generation is readily accomplished by humans (and now, LLMs), language identification in the Gold–Angluin model proves intractable for virtually all non-trivial collections $\mathcal{L}$ – even collections of regular languages, which are considerably simpler than human languages. This intractability persists despite Gold's model imposing no constraints on the computational power of the learner.

Kleinberg and Mullainathan [2024] offer a striking resolution to this paradox: they demonstrated that a subtle shift in the problem formulation – from identification to generation – renders the problem tractable. Their main result is that language generation in the limit is achievable for *any* countable collection of languages. This remarkable finding sparked significant interest within the learning theory community, spawning a growing line of works (*e.g.*, Li et al. [2025], Kalavasis et al. [2025], Charikar and Pabbaraju [2025], Raman and Raman [2025]); see Section 1.2 for a detailed discussion.

Most relevant to our work is the work of Li et al. [2025] – who take a learning theory perspective on language generation in the limit (henceforth, simply language generation) – allowing $\mathcal{L}$ to be an uncountable collection – and characterizing which collections $\mathcal{L}$ are amenable to different forms of generation. In particular, they define a hierarchy of generation notions which differ in whether the number of samples $n^\star$ that are needed to achieve consistent generation depends on the target language $K \in \mathcal{L}$ or its enumeration. In decreasing order of hardness, these notions are:

▷ **Uniform Generation** where $n^\star$ neither depends on $K$ nor its enumeration; it only depends on $\mathcal{L}$

▷ **Non-uniform Generation** where $n^\star$ can depend on $\mathcal{L}$ and $K$ but not $K$'s enumeration

▷ **Generation** where $n^\star$ can depend on $\mathcal{L}$, $K$, and $K$'s enumeration

Here, the weakest notion, generation, is the one studied by Kleinberg and Mullainathan [2024].

**Our Main Questions.** In their work, Li et al. [2025] demonstrated that the problem of generation, under any of these notions, is fundamentally distinct from prediction and the PAC learning framework: they constructed several classes that are generatable but not PAC learnable and vice versa, establishing a clear separation between these paradigms. While these examples illustrate specific differences between prediction and generation, they raise a more profound question: do the conceptual properties of generation differ significantly from those of prediction? One of the most fundamental properties of traditional prediction tasks (such as binary, multiclass, or online classification) is closure under finite unions – if classes $\mathcal{H}_1$ and $\mathcal{H}_2$ can be learned (under any of the aforementioned notions of prediction) then so can $\mathcal{H}_1 \cup \mathcal{H}_2$. However, establishing the closure of generatability has remained elusive so far. Indeed, the challenge of effectively combining generators also appears in the work of Kalavasis et al. [2025], where it is a crucial obstacle in establishing tight sample complexity bounds for generation in statistical settings. Although Kalavasis et al. [2025] ultimately circumvent this challenge through alternative techniques, the following fundamental question remains:

▷ **Q1.** *Is generation closed under finite unions?*

This was also explicitly posed as a key open problem by Li et al. [2025] (as Questions 6.2) and as a first step in addressing this broader question, they also posed the following more specific variant:[3]

▷ **Q2.** *Are finite unions of non-uniformly generatable classes always generatable?*

Apart from differences between generation and prediction, two main results of Li et al. [2025] are complete characterizations of uniform generation and non-uniform generation, respectively. A natural and important question is to develop a complete characterization of generation in the limit. Toward this goal, Li et al. [2025] provide several sufficient conditions. Perhaps the most natural one of these is based on the Eventually Unbounded Closure (EUC) property (see Definition 2.7). Informally, a

---

[3]Li et al. [2025] show finite unions of uniformly generatable classes are generatable but need not be non-uniformly generatable.

collection $\mathcal{L}$ satisfies the EUC property if for any target language, $K \in \mathcal{L}$, after seeing a finite number of elements $x_1, x_2, \ldots, x_n \in K$, all languages $L \in \mathcal{L}$ consistent with $x_1, x_2, \ldots, x_n$ share infinitely many elements. (In other words, all languages in the "version space" defined by $x_1, x_2, \ldots, x_n$ share infinitely many elements for large enough $n$.) Further, Li et al. [2025] observed that the EUC property is connected to a certain "autoregressive" property where, after seeing finitely many samples, the generator no longer needs to observe further samples to generate new and unseen examples in the future. Given the central role of EUC in current approaches to characterizing generatability and its connection to the autoregressive property, Li et al. [2025] pose the following question:

▷ **Q3.** *Is there a non-uniformly generatable and uncountable class violating the EUC property?*

## 1.1 Our Contributions

We present a family of constructions and corresponding techniques that resolve three open questions of Li et al. [2025] in the model of language generation of Kleinberg and Mullainathan [2024].

▷ **Non-Closure Under Finite Unions:** Our first result constructs two collections $\mathcal{L}_1$ and $\mathcal{L}_2$ that are generatable individually, while their union $\mathcal{L}_1 \cup \mathcal{L}_2$ is not, answering **Question 1** negatively (Theorem 3.1). This construction relies on specific properties relating the two collections (which we explain later; Section 3.4), allowing us to develop a family of counterexamples – demonstrating that the failure of union-closedness is not isolated but an inherent property of language generation that fundamentally distinguishes it from traditional prediction tasks. In particular, we can also ensure that $\mathcal{L}_1$ and $\mathcal{L}_2$ are both non-uniformly generatable, hence, also resolving **Question 2**.

▷ **A Minimal Pair of Classes Whose Union Is Not Generatable:** Next, we investigate the spectrum between known extremes: at one end, Kleinberg and Mullainathan [2024] showed that unions of countable collections are generatable; at the other end, our first result (Theorem 3.2) shows that unions of uncountable generatable collections need not be generatable. Our second result, refines this understanding by constructing a countable collection $\mathcal{L}_1'$ and an uncountable collection $\mathcal{L}_2'$ whose union is not generatable. Notably, in this construction, $\mathcal{L}_1'$ is non-uniformly generatable and $\mathcal{L}_2'$ is uniformly generatable (both without requiring to get any examples from the target language), yet their union $\mathcal{L}_1' \cup \mathcal{L}_2'$ is not generatable (Theorem 3.2). This observation shows that the construction is minimal in the sense that simplifying either collection further (by, *e.g.*, making $\mathcal{L}_1'$ to be uniformly generatable or $\mathcal{L}_2'$ countable) would make the union generatable, due to results of Li et al. [2025], Kleinberg and Mullainathan [2024].

▷ **Non-Uniformly Generatable Collections without EUC:** Our third result (Theorem 3.3) answers **Question 3** by identifying an uncountable collection that is non-uniformly generatable but violates the Eventually Unbounded Closure (EUC) property. This result provides insight into the relationship between non-uniform generatability and the autoregressive property, suggesting that current sufficient conditions for characterizing generation in the limit need to be expanded.

**Technical Novelty.** We now highlight the key technical challenges and innovations in obtaining our results. The problem of studying generatability of unions of classes was previously examined by Li et al. [2025], who showed that countable unions of uniformly generatable classes need not be generatable in the limit. As they note, their result "showcases the hardness of characterizing generatability in the limit." Their construction is highly sophisticated, involving a countable list of collections $\mathcal{L}_1, \mathcal{L}_2, \ldots$, each defined by a different prime number. In this construction, they are able to ensure that the union, $\bigcup_i \mathcal{L}_i$, was sufficiently complex that it ends up being non-generatable. This is achievable because they take unions of countably many classes – an operation under which even standard prediction tasks in learning theory (such as binary and multi-class classification) are not closed.[4]

A key challenge in our work is that we consider unions of just two classes. These two classes must simultaneously possess enough structure to be individually generatable, yet become sufficiently complex when combined that their union is not generatable in the limit. Our result in Theorem 3.2 is even more surprising, as it demonstrates this phenomenon with one class being (trivially) uniformly generatable and the other being countable. Our construction relies on several careful choices that we outline below and elaborate on in Section 3.4.

*Overview of Diagonalization.* The only method for proving non-generatability is *diagonalization* [Li et al., 2025, Charikar and Pabbaraju, 2025, Kalavasis et al., 2024]. This approach is

---

[4]Indeed, one can construct a countable union of classes where the $i$-th class has VC dimension $i$, so that the countable union necessarily has infinite VC dimension, rendering it unlearnable.

also fundamental in computational complexity theory [Arora and Barak, 2009]. At a very high level, using diagonalization one constructs, for any consistent generator, an adversarial enumeration of the target language with distinct "phases" $t_1, t_2, \ldots$ such that in the $i$-th phase, the generator must generate from language $L_i$ (and not from languages $L_{i+1}, L_{i+2}, \ldots, L_\infty = K$). Consequently, either the generator fails to be consistent in one phase (failing to generate from $L_i$), or we identify infinite steps where it fails to generate from $K = L_\infty$.

*Challenges in Using Diagonalization with Finite Unions.* When applying this argument to two collections, we must partition the languages $L_1, L_2, \ldots$ between our collections, $\mathcal{L}_1$ and $\mathcal{L}_2$. However, this causes a problem: by the Pigeonhole principle, at least one collection must contain infinitely many of these languages. Typically, this would allow us to reproduce a similar diagonalization argument, showing that the corresponding collection (say $\mathcal{L}_1$) is itself not generatable – contradicting our goal. (Note that Li et al. [2025] avoid this problem by using countably many collections, assigning language $L_i$ to collection $\mathcal{L}_i$ – ensuring that each collection only has a single language from $L_1, L_2, \ldots$.)

*Idea 1 (Embedding A Shared Structure).* To overcome this challenge, we divide the languages into two sets $L_1, L_3, L_5, \ldots$ and $L_2, L_4, L_6, \ldots$, embed a shared structure across each set of languages, and assign the sets to the collections $\mathcal{L}_1 = \{L_1, L_3, L_5, \ldots\}$ and $\mathcal{L}_2 = \{L_2, L_4, L_6, \ldots\}$ respectively. For instance, one form of structure is to ensure that $L_1, L_3, L_5, \ldots \supseteq \mathbb{Z}_+$ (all odd-indexed languages contain all positive integers) and $L_2, L_4, L_6, \ldots \supseteq \mathbb{Z}_-$ (all even-indexed languages contain all negative integers). This structure prevents the diagonalization argument from working independently on either $\mathcal{L}_1$ or $\mathcal{L}_2$.

This approach, however, introduces a new challenge: it potentially prevents diagonalization from working on $\mathcal{L}_1 \cup \mathcal{L}_2$ as well. For example, a generator could determine whether it is in the "even world" (where $K \supseteq \mathbb{Z}_-$) or the "odd world" (where $K \supseteq \mathbb{Z}_+$) by comparing the length of observed prefixes of negative integers versus positive integers. (We make this argument formal in Section 3.4.)

*Idea 2 (Omitting Shared Elements).* To prevent such identification, we need to ensure that after seeing only finitely many elements, every even-indexed language consistent with the enumerated stream is sufficiently similar to some odd-indexed language, and vice-versa. In the context of our running example, to do that we allow each language $L_i$ to omit finitely many elements from its shared set (either $\mathbb{Z}_+$ or $\mathbb{Z}_-$). This turns out to be sufficient to ensure that the generator cannot identify whether $K \in \mathcal{L}_1$ or $K \in \mathcal{L}_2$.

This idea, however, creates an additional technical hurdle because standard diagonalization leads to the language $K (= L_\infty)$ missing infinitely many elements from both shared sets (positive and negative integers for $\mathcal{L}_1$ and $\mathcal{L}_2$ respectively), meaning it could not belong to either collection, otherwise the corresponding collection would be non-generatable.

*Idea 3 (A Variant of Diagonalization).* Our third innovation is a novel form of diagonalization where, counter-intuitively, the adversary forces the generator to make mistakes only in alternate rounds. This approach carefully balances the constraints to ensure that (1) both collections remain generatable individually, (2) their union is not generatable, and (3) the final language $K$ enumerated during diagonalization belongs to one of the collections.

We believe this novel diagonalization technique will have broader applications in the study of language generation, potentially paving the way toward a complete characterization of generatability in the limit – addressing the main open question in this research area identified by Li et al. [2025].

## 1.2 Related Work

Our work directly builds on the framework of Kleinberg and Mullainathan [2024], who introduced the model of language generation in the limit. Since then, a growing line of research has explored various aspects of language generation (*e.g.*, Li et al. [2025], Kalavasis et al. [2025], Charikar and Pabbaraju [2025], Raman and Raman [2025], Peale et al. [2025], Kleinberg and Wei [2025]). Here, we discuss the most relevant prior works.

**Uniform and Non-Uniform Generation.** Li et al. [2025] introduced a hierarchy of three notions of generation – uniform, non-uniform, and generatable – and provided characterizations for uniform and non-uniform generation along with sufficient conditions for generatability. Charikar and Pabbaraju [2025] also, independently and concurrently, studied non-uniform generation and showed that all countable collections can be non-uniformly generated, strengthening the results of Kleinberg

and Mullainathan [2024] (which only showed that countable collections are generatable and finite collections are uniformly generatable).

**Language Generation with Breadth.** While Kleinberg and Mullainathan's algorithm eventually ceases outputting elements outside of $K$ after finite time (*i.e.*, it eventually stops hallucinating), this property comes at a cost: the algorithm sacrifices *breadth – i.e.*, the ability to generate diverse strings from the target language. A number of works study language generation with different notions of breadth and demonstrate that requiring many natural notions of breadth makes generation significantly harder, almost as hard as language identification Kalavasis et al. [2025], Charikar and Pabbaraju [2025], Kalavasis et al. [2024], Peale et al. [2025], Kleinberg and Wei [2025].

**Further work on Language Generation.** Recent works have also explored several other aspects of language generation. Raman and Raman [2025] investigated language generation in a model where an adversary can introduce errors in the inputs, developing a robust framework for noisy settings. Karbasi et al. [2025] explored the complexity of determining if a specific generator $\mathcal{G}$ is hallucinating.

**Union-Closedness of Prediction.** Understanding the behavior of learning problems such as binary classification and online learning under various natural operations on the underlying hypothesis class, *e.g.*, finite unions, intersections, and products, is a fundamental challenge that is now well-understood in the literature – see, *e.g.*, Van Der Vaart and Wellner [2009], Alon et al. [2020], Ghazi et al. [2021] and references therein. Notably, these properties also reveal natural learning strategies: one can decompose the underlying class into simpler ones, learn them separately, and "combine" the learners.

## 2 Preliminaries

In this section, we present some background on language generation in the limit.

**Notation.** Let $\Sigma$ be a finite alphabet (*e.g.*, $\{a, b, \ldots, z\}$), and $\Sigma^*$ the set of all finite-length strings formed by concatenating symbols from $\Sigma$. We define a language $L$ as an infinite subset of $\Sigma^*$. A collection of languages is denoted by $\mathcal{L}$. We define a generating algorithm $\mathcal{G} = (\mathcal{G}_n)_{n \in \mathbb{N}}$ as a sequence of mappings $\mathcal{G}_n \colon (\Sigma^*)^n \to \Sigma^*$ parametrized by the input size $n$. In words, the generator maps a finite training set to a (potentially infinite) set of elements.[5]

**Language Generation in the Limit.** We begin with the formal definition of language generation in the limit, introduced by Kleinberg and Mullainathan [2024].

**Definition 2.1** (Language Generation in the Limit [Kleinberg and Mullainathan, 2024]). *Fix some $K$ from the language collection $\mathcal{L}$ and a generating algorithm $\mathcal{G} = (\mathcal{G}_n)$. At each step $n$, let $S_n \subseteq K$ be the set of all strings that the algorithm $\mathcal{G}$ has seen so far. $\mathcal{G}$ must output a string $w_n \notin S_n$ (its guess for an unseen string in $K$). The algorithm $\mathcal{G}$ is said to generate from $K$ in the limit if, for all enumerations of $K$, there is some $n^* \in \mathbb{N}$ such that for all steps $n \geq n^*$, the algorithm's guess $w_n$ belongs to $K \setminus S_n$ (or $K \setminus S_n$ is empty). The collection $\mathcal{L}$ allows for generation in the limit if there is an algorithm $\mathcal{G}$ that generates from $K$ in the limit for any $K \in \mathcal{L}$.*

To gain some intuition about this definition, consider the collection $\mathcal{L} = \{\mathbb{Z}, L_1, L_{-1}, L_2, L_{-2}, \ldots\}$ of thresholds over integers where, for each $i \in \mathbb{Z}$, $L_i = \{i, i+1, i+2, \ldots\}$. Suppose the target language is some $K \in \mathcal{L}$ and the adversary first enumerates string $x_1$. The generator can deduce that $K = L_z$ for some $z \leq x_1$, *i.e.*, $K \in \{\mathbb{Z}, L_{x_1}, L_{x_1-1}, \ldots\}$. Since the intersection of all of these languages is non-empty and is a strict superset of the strings enumerated so far (namely, the intersection is $\{x_1 + 1, x_1 + 2, \ldots\}$), the generator can generate an element that is guaranteed to be in $K$: for instance, it is sufficient to output $x_1 + 1$. More generally, after seeing strings $x_1, x_2, \ldots, x_i$, the generator can output any integer larger than $\max\{x_1, x_2, \ldots, x_i\}$.

*Remark* 2.2. For the problem to be interesting, we assume that each language in the collection has infinite cardinality, *i.e.*, $|L| = \infty$ for all $L \in \mathcal{L}$. (Otherwise, $K \setminus S_n$ eventually becomes empty.)

*Remark* 2.3 (Repetitions). For simplicity, we throughout also assume that the adversary is not allowed to repeat strings in its enumeration. Otherwise, to define uniform and non-uniform generatability, we have to count the number of unique elements listed by the adversary [Li et al., 2025].

---

[5]While Kleinberg and Mullainathan [2024] required outputting only one element at a time, Kalavasis et al. [2025], Charikar and Pabbaraju [2025] relaxed this to allow for case where, at some finite point, one can stop training and generate a rich set of responses.

Kleinberg and Mullainathan [2024] showed that language generation in the limit is possible for all countable collections of languages – starkly contrasting results in language identification.

**Uniform and Non-Uniform Generation in the Limit.**  Next, we present two strengthenings of the notion of generation in the limit introduced by Li et al. [2025]. The first notion is the strongest, it requires the number $n^*$ in Definition 2.1 (which is the number of samples required by the generator before it starts generating consistently) to be independent of the target language and its enumeration.

**Definition 2.4** (Uniform Generation [Li et al., 2025]). *A collection $\mathcal{L}$ is said to be uniformly generatable, if there is an algorithm $\mathcal{G}$ and $n^*$ such that, for each $K \in \mathcal{L}$ and each (adversarially chosen) enumeration $E_K$ of $K$, $\mathcal{G}$ generates from $K$ in the limit after seeing $n \geq n^*$ examples from $K$.*

The non-uniform generation weakens this notion by allowing $n^*$ to depend on the target language.

**Definition 2.5** (Non-Uniform Generation [Li et al., 2025]). *A collection $\mathcal{L}$ is said to be non-uniformly generatable, if there is an algorithm $\mathcal{G}$ such that, for each $K \in \mathcal{L}$, there is a number $n^* = n^*(K)$ such that for any (adversarially chosen) enumeration $E$ of $K$, $\mathcal{G}$ generates from $K$ in the limit after seeing $n \geq n^*$ examples from $K$.*

Li et al. [2025] provide characterizations for both the collections that are uniformly and non-uniformly generatable, respectively. These, in particular, show that all countable collections are non-uniformly generatable; a result independently and concurrently also shown by Charikar and Pabbaraju [2025]. They also show that all finite collections are uniformly generatable in the limit; a result that was also earlier shown by Kleinberg and Mullainathan [2024].

A key remaining question, after these works, is a characterization for the weakest notion of generation in the limit in Definition 2.1; where $n^*$ can depend on both $K$ and $E_K$. Next, we define the eventually unbounded closure property, which forms the basis of a sufficient condition for generatability proposed by Li et al. [2025].

**Definition 2.6** (Version Space). *Given a finite sequence of strings $X = \{x_1, \ldots, x_n\}$ and a collection $\mathcal{L}$, let $V(\mathcal{L}, X)$ be the set of languages $L \in \mathcal{L}$ containing $X$.*

**Definition 2.7** (Eventually Unbounded Closure). *A collection $L$ is said to have the Eventually Unbounded Closure (EUC) property if, for every $L \in \mathcal{L}$ and enumeration $x_1, x_2, \ldots$ of $L$, there is a finite time $t$, at which all languages in $V(\mathcal{L}, \{x_1, x_2, \ldots, x_t\})$ share infinitely many elements.*

It is not too hard to show that all uniformly generatable collections possess the EUC property, while some generatable collections do not [Li et al., 2025]. The relationship between EUC and non-uniform generatability is less clear, and Li et al. [2025] constructed a countable class that was non-uniformly generatable without the EUC property, but left open whether an uncountable such class exists. Our work resolves this question by demonstrating an uncountable class that is non-uniformly generatable yet lacks the EUC property.

# 3  Our Results and Technical Overview

In this section, we present our results on language generation in the limit, addressing three open questions of Li et al. [2025].

## 3.1  Non-Closure Under Finite Unions

Our first result demonstrates that generation is not closed under finite unions.

**Theorem 3.1.** *There are uncountable collections $\mathcal{L}_1, \mathcal{L}_2$ that are non-uniformly generatable while $\mathcal{L}_1 \cup \mathcal{L}_2$ is not generatable.*

We stress that while each collection is non-uniformly generatable, their union does not just violate non-uniform generatability, it also violates generatability. Thus, Theorem 3.1 resolves both Questions 1 and 2 negatively. Further, the construction in Theorem 3.1 relies on a certain "prefix-realizability" properties between $\mathcal{L}_1$ and $\mathcal{L}_2$ (see Appendix A.2), which enable us to generalize this approach to a family of counterexamples. This demonstrates that the failure of union-closedness is not an isolated phenomenon but rather an inherent property of language generation.

## 3.2  On Finite Union of Non-Uniformly Generatable Collections

Both collections in Theorem 3.1 are uncountable. This is necessary in part: at least one collection must be uncountable since, otherwise, if both are countable collections, then their union remains

countable and, hence, generatable by the results of Kleinberg and Mullainathan [2024]. This naturally raises the question: *must both collections be uncountable for the union to be non-generatable?*

Our second result shows that both collections need not be uncountable. From our family of counterexamples, we identify a pair $(\mathcal{L}_1, \mathcal{L}_2)$ where $\mathcal{L}_1$ is countable, $\mathcal{L}_2$ is uncountable, and their union is not generatable.

**Theorem 3.2.** *There are collections $\mathcal{L}_1$ and $\mathcal{L}_2$ for which $\mathcal{L}_1 \cup \mathcal{L}_2$ is not generatable such that:*

   ▷ $\mathcal{L}_1$ *is countable and non-uniformly generatable, without requiring any elements from the adversary*

   ▷ $\mathcal{L}_2$ *is uncountable and uniformly generatable, without requiring any elements from the adversary.*

Thus, Theorem 3.2 establishes that unions of non-uniformly and uniformly generatable classes are not guaranteed to be generatable. Moreover, these collections are minimally complex in the following precise sense: if either collection is any simpler (*i.e.*, if $\mathcal{L}_1$ is uniformly generatable or $\mathcal{L}_2$ is countable), then $\mathcal{L}_1 \cup \mathcal{L}_2$ would be generatable. When $\mathcal{L}_1$ is uniformly generatable this is due to a result of Li et al. [2025] which shows that unions of uniformly generatable classes are generatable. When $\mathcal{L}_2$ is countable, this is due to a result of Kleinberg and Mullainathan [2024] showing that all countable collections are generatable.

Furthermore, even among non-uniformly generatable and uniformly generatable classes, $\mathcal{L}_1$ and $\mathcal{L}_2$ represent the simplest collections because they can be generated without observing any elements from the adversary. In the terminology of Li et al. [2025], this enables "autoregressive" generation – where the generator can produce new elements without requiring input from the adversary.

Finally, since $\mathcal{L}_1$ is countable it can be expressed as a union of singletons $\mathcal{L}_1 = \bigcup_{i=1}^{\infty} \{L_i\}$. Since each singleton is trivially uniformly generatable, Theorem 3.2 provides a list of countably many classes $(\mathcal{L}_2, \{L_1\}, \{L_2\}, \dots)$ that are each uniformly generatable without requiring any elements from the adversary but their union $\mathcal{L}_2 \cup \{L_1\} \cup \{L_2\} \cup \dots$ is not generatable. This, in particular, recovers a result of Li et al. [2025], namely their Lemma 4.3.

### 3.3 On the EUC Property and Non-Uniform Generation

Our third result resolves Question 3 as a direct consequence of our construction in Theorem 3.1.

**Theorem 3.3.** *There is an uncountable collection $\mathcal{L}$ that is non-uniformly generatable and violates the EUC property (Definition 2.7).*

Specifically, the collection $\mathcal{L}_1$ from Theorem 3.1 is non-uniformly generatable but violates the EUC property, providing a concrete counterexample that addresses this open question.

### 3.4 Technical Overview

In this section, we give a detailed overview of our approach; for a higher-level summary, we refer readers to Section 1.1. Our goal is to show that finite unions of generatable classes need not be generatable. Toward this, it is instructive to build some intuition about how to show that a class is *not* generatable. Recall that Kleinberg and Mullainathan [2024] showed that *any* countable collection is generatable, thus we must necessarily work with uncountable collections.

**Warm-Up: A Non-generatable Collection.** Let the domain $\Sigma^*$ be $\mathbb{N}$. A natural candidate to consider is the collection of all *infinite* subsets of $\mathbb{N}$.[6] Given a generator $\mathcal{G}$, we will pick some $K \in \mathcal{L}$ and enumeration of $K$, both tailored to $\mathcal{G}$, so that it makes a mistake in *every* step. Let us denote the element enumerated at step $i \in \mathbb{N}$ by $x_i$ and the element generated at step $i \in \mathbb{N}$ by $y_i = \mathcal{G}_i(x_1, \dots, x_i)$. We begin by setting $x_1 = 1$. Then, for each $k \in \mathbb{N}$ we define $x_k = \max\{\max_{i=1,\dots,k-1} x_i, \max_{j=1,\dots,k-1} y_j\} + 1$. We let $K = \bigcup_{i \in \mathbb{N}} \{x_i\}$ and $E = (x_1, x_2, \dots)$. Notice that $x_i \neq x_{i'}$ for $i \neq i'$, hence $K$ contains infinitely many elements. Moreover, $K \in \mathcal{L}$ and $E$ is a valid enumeration of $K$.[7] Why does the learner make a mistake in every step? Consider two cases: at step $i$ either the learner outputs some $y_i \in \{x_1, \dots, x_i\}$ so it fails to output an unseen element, or it outputs some $y_i \notin \{x_1, \dots, x_i\}$. In the latter case, by the definition of $E$, all the elements we

---

[6]Recall that for the problem of generation to be well-defined Kleinberg and Mullainathan [2024] require that all languages in the collection are infinite.

[7]In fact, it is an "easy" enumeration since it lists elements in increasing order.

enumerate in subsequent rounds (the unseen elements of $K$), are greater than $y_i$, which means that $y_i$ is not an unseen element of $K$.

Unfortunately, the collection $\mathcal{L}$ is very complex so it is not clear at all if it is possible to split it into two collections $\mathcal{L}_1, \mathcal{L}_2$ such that both of them are generatable and $\mathcal{L}_1 \cup \mathcal{L}_2 = \mathcal{L}$. Nevertheless, this idea of constructing hard enumerations and target languages as a function of the underlying generator – a.k.a. diagonalizing against the underlying generator – will be the first ingredient towards our main result. Next, we explore the crucial question: Can we construct $\mathcal{L}_1, \mathcal{L}_2$ that are individually easy to generate from but $\mathcal{L}_1 \cup \mathcal{L}_2$ is (almost) as complex as the previous collection?

**A First Attempt (which Fails).** Our first idea is to create $\mathcal{L}_1$ and $\mathcal{L}_2$ in a symmetric way. Both defined over $\mathbb{Z}$, $\mathcal{L}_1$ contains languages that are "easy" on the negative integers and "hard" on the positive integers, while $\mathcal{L}_2$ contains languages that are "hard" on the negative integers and "easy" on the positive integers. Taking this approach to the extreme, we define

$$\mathcal{L}_1 := \{L_A := \mathbb{Z}_- \cup A, \ A \subseteq \mathbb{Z}_+, \ |A| = \infty\} \quad \text{and} \quad \mathcal{L}_2 := \{L_B := \mathbb{Z}_+ \cup B, \ B \subseteq \mathbb{Z}_-, \ |B| = \infty\} .$$

In words, every language in $\mathcal{L}_1$ contains all the negative integers and some infinite subset of the positive integers (and for every such subset there exists a corresponding language in $\mathcal{L}_1$). The collection $\mathcal{L}_2$ is defined in a symmetric way where the roles of positive and negative integers are flipped. Both $\mathcal{L}_1$ and $\mathcal{L}_2$ are uniformly generatable in a trivial way: for $\mathcal{L}_1$ it suffices to generate *any* unseen negative number and symmetrically for $\mathcal{L}_2$.

Can a single generator successfully generate $\mathcal{L}_1 \cup \mathcal{L}_2$? Since there is no systematic way to generate from uncountable collections (unlike countable collections), we explore natural heuristics. Perhaps the most intuitive approach is to track the positive and negative integers observed so far and generate from the "heavier" side (the side with more observed elements). This approach fails: an adversary can select a language from $\mathcal{L}_1$ and ensure any finite prefix of the enumeration contains more positive than negative integers – fooling the learner into generative positive integers – and, then, selecting the set $A$ to using our warm-up technique to force mistakes. However, a more sophisticated generator does exist for $\mathcal{L}_1 \cup \mathcal{L}_2$: the generator keeps track of the longest prefix[8] enumerated on the positive and negative side separately, and outputs an unseen number from the side with the longer prefix.

Why does this work? If $K = \mathbb{Z}$, then the generator is always correct. Otherwise, $K$ either contains $\mathbb{Z}_-$ and misses at least one element from $\mathbb{Z}_+$ or it contains $\mathbb{Z}_+$ and misses at least one element from $\mathbb{Z}_-$. In either case, since the adversary must completely enumerate $K$ the prefix of one of the two sides will stop growing, while the other increases indefinitely. Thus, in the limit, this generator will output valid and unseen elements from $K$.

**An Attempt (which Works).** The core reason why our previous attempt fails is that (in the limit), a sufficiently clever generator can identify if $K \in \mathcal{L}_1$ or $K \in \mathcal{L}_2$. To get our result, we need to ensure that no generator can make this determination. We therefore modify our collections $\mathcal{L}_1, \mathcal{L}_2$ to make them even more similar while keeping them individually generatable:

$$\mathcal{L}'_1 := \{L_{A,B} := (\mathbb{Z}_- \setminus A) \cup (\mathbb{Z}_+ \setminus B), A \subseteq \mathbb{Z}_-, |A| < \infty, B \subseteq \mathbb{Z}_+, |B| = \infty\} ,$$
$$\mathcal{L}'_2 := \{L_{A,B} := (\mathbb{Z}_- \setminus A) \cup (\mathbb{Z}_+ \setminus B), A \subseteq \mathbb{Z}_-, |A| = \infty, B \subseteq \mathbb{Z}_+, |B| < \infty\} .$$

In this construction, languages in $\mathcal{L}'_1$ contain almost all negative integers (missing only finitely many in $A$) and only some positive integers (missing infinitely many in $B$). The collection $\mathcal{L}'_2$ is defined symmetrically, with finite/infinite exclusions reversed. First, notice that $\mathcal{L}'_1, \mathcal{L}'_2$ are non-uniformly generatable in a trivial way: there exists a generator that generates from $\mathcal{L}_1$ (and $\mathcal{L}'_2$) without seeing any examples from the target language and the number of mistakes it makes depends only on $K$.[9]

The crux of the difficulty with the new collections is that, while they have a clear asymmetry (one side missing finitely many elements, the other missing infinitely many), this asymmetry cannot be detected at any finite time. Indeed, it is not hard to see that the sophisticated generator that kept track of continuous prefixes of both positive and negative integers and generated according to the longer-prefix side fails for $\mathcal{L}'_1 \cup \mathcal{L}'_2$. It turns out that *every* generator fails for the union of this pair of collections.

To show this, a tempting approach is to try to replicate our strategy from the warm-up construction and force a mistake in *every* round. For instance, we can start enumerating positive integers, and if

---

[8] A prefix of length $i$ from $\mathbb{Z}_-$ is the set $\{-i, -i-1, \ldots, -1\}$ and symmetrically from $\mathbb{Z}_+$

[9] Indeed, for $\mathcal{L}'_1$ the generator can start outputting negative integers in decreasing order and since any $K$ misses only finitely many of them it will eventually start generating correctly (symmetrically for $\mathcal{L}'_2$.)

the generator ever outputs an unseen positive number $x_{i_1}$, switch to enumerating negative integers ensuring that we do not enumerate any negative number the generator has generated. Indeed, $\mathcal{L}'_1 \cup \mathcal{L}'_2$ is sufficiently complex to ensure that for every $t \in \mathbb{N}$ the elements $S_t$ enumerated up to timestep $t$ are consistent with some language $K_t \in \mathcal{L}$. Unfortunately, this property does *not* imply that we present a complete enumeration of some $K \in \mathcal{L}$. Indeed, it is not hard to construct generators for which the stated approach ends up enumerating some $\widehat{K}$ that does not contain infinitely many negative integers and infinitely many positive integers; such languages are not part of $\mathcal{L}'_1 \cup \mathcal{L}'_2$. Hence, to show the non-generatability of $\mathcal{L}'_1 \cup \mathcal{L}'_2$ we must use a more involved argument.

**A Modified Diagonalization Argument.** We now present a lower bound construction that proceeds in (potentially infinitely) many phases, which are now broken into two subphases, and is tailored to the underlying generator $\mathcal{G}$. Let us introduce some notation: let $S_t$ denote the elements enumerated by the adversary up to (and including) step $t$. We also denote by $P_t$ the set of *positive* integers the generator has outputted up to (and including) step $t$ and $N_t$ set of *negative* integers the adversary has enumerated up to (and including) step $t$. We now describe our construction inductively:

Phase 1-A: During this phase, the adversary enumerates the negative integers starting from $-1$ in a sequential, decreasing order $(-1, -2, -3, \dots)$ until the generator outputs an unseen negative integer. If this never happens, the adversary ends up enumerating $K = \mathbb{Z}_- \in \mathcal{L}'_1$ and the generator makes a mistake in every timestep, hence the adversary wins the game. Thus, let us assume that $t_{1,A}$ is the first timestep the generator outputs an unseen negative integer. At this point, we switch to phase 1-B.

Phase 1-B: The adversary now enumerates positive integers starting from $\max P_{t_{1,A}} + 1$ in a sequential increasing order. Let $t_{1,B} > t_{1,A}$ the first timestep when the generator outputs a positive integer. Using similar reasoning, if this never happens, the adversary enumerates a valid language from $\mathcal{L}'_2$ and the generator makes infinitely many mistakes. Otherwise, we proceed to phase 2.

For $\ell \geq 2$, the $\ell$-th phase follows:

Phase $\ell$-A: Upon entering this phase, the adversary enumerates the largest negative integer not yet enumerated, *i.e.*, the number $\min N_t - 1$, and in the subsequent rounds of this phase it continues in decreasing order. Importantly, some of these negative integers might coincide with elements the generator has previously outputted, thus "correcting" some of its tentative mistakes. As we explained, this is unavoidable since there are $\mathcal{G}$ such that if the adversary forces $\mathcal{G}$ to make a mistake in every step it necessarily enumerates a language that is not in $\mathcal{L}'_1 \cup \mathcal{L}'_2$. Let $t_{\ell,A} > t_{\ell-1,B}$ be the first timestep when the generator outputs an unseen negative integer. At this point, we move to subphase $\ell$-B.

Phase $\ell$-B: Similar to subphase 1-B, the adversary enumerates positive integers starting from $\max P_{t_{\ell,A}} + 1$ until the generator outputs a positive integer, at which point this subphase ends.

Having described the construction, we now argue its correctness. As we argued already, if the construction terminates after finitely many phases, then the learner makes infinitely many mistakes and the adversary enumerates a valid language. Alternatively if the construction proceeds for infinitely many phases, then the adversary enumerates $\mathbb{Z}_-$ along with an infinite subset of $\mathbb{Z}_+$, producing a valid language from $\mathcal{L}'_1$. What about the generator's mistakes? The crucial observation is that every time we move from subphase $\ell$-B to $(\ell + 1)$-A the adversary forces a mistake for the generator. With infinitely many phases, the generator necessarily makes infinitely many mistakes. The formal details are in Appendix B.1.

*Remark* 3.4 (Connection to EUC). This construction yields an additional insight: there are uncountable classes that are non-uniformly generatable (in a trivial way), yet do not satisfy the Eventually Unbounded Closure (EUC) property. Indeed, both $\mathcal{L}'_1$ and $\mathcal{L}'_2$ are uncountable and non-uniformly generatable, but we can prove they violate EUC (Appendix B.2), thus addressing Question 3.

*Remark* 3.5 (Stronger Lower Bound). The ideas we described in this section can be utilized to derive the stronger lower bound from Theorem 3.2. The formal details are in Appendix B.2.

## 4 Concluding Remarks

In this paper we have continued studying the emerging line of work on language generation in the limit, resolving three open questions from Li et al. [2025]. While the work of Kleinberg and Mullainathan [2024] showed a remarkable tractability of the problem when the collection of languages

is *countable*, our results and the results of Li et al. [2025] show that this learning task is significantly more involved when we move to uncountable collections and exhibits behaviors that are qualitatively different from traditional learning problems such as binary classification [Valiant, 1984, Vapnik and Chervonenkis, 2015] or online learning [Littlestone, 1988]. Our results further highlight the difficulty of coming up with a *characterization* of generatibility, since there are collections that are generatable (in a strong sense), yet even taking the union of two of them yields a non-generatable collection. Nevertheless, we hope that the technique we introduce to show non-generatability will help pave the way to the solution of this challenging problem.

**Acknowledgments**

This research was supported (in part) by the AI Institute for Learning-enabled Optimization at Scale (TILOS).

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

# A  Further Results

## A.1  Formal Lower Bound of Non-Generatability

In this section we show formally that the class of all infinite languages over $\mathbb{N}$ is not generatable in the limit. It is worth mentioning that this result follows from a result of Li et al. [2025]. Here, we give a simpler proof which also generalizes to *randomized* learners (Proposition A.2).

**Proposition A.1.** *Let $\Sigma^* = \mathbb{N}$ and $\mathcal{L}$ be the set of all infinite subsets of $\mathbb{N}$. Then, no deterministic algorithm can generate from $\mathcal{L}$ in the limit.*

*Proof.* Let $\mathcal{G} = (\mathcal{G}_n \colon \mathbb{N}^n \to \mathbb{N})_{n \in \mathbb{N}}$ be a deterministic generating algorithm and, for each $n$, let $\mathcal{G}_n(x_1, x_2, \ldots, x_n)$ be the string outputted by $\mathcal{G}_n$ when provided the strings $x_1, x_2, \ldots, x_n$ as input. We will show that there is a language $K \in \mathcal{L}$ and an enumeration $E$ of $K$, both of which depend on $\mathcal{G}$, for which $\mathcal{G}$ fails to generate from $K$ in the limit. We will construct the enumeration and the target language inductively. For all $n \in \mathbb{N}$, we denote by $E_n$ the first $n$ elements of $E$.

Let $x_1 = 1$ and $E_1 = (1)$. We then define $x_2 = \max \{x_1, \mathcal{G}_1(x_1)\} + 1$ and $E_2 = (x_1, x_2)$. Continuing in the same fashion, for any $n \in \mathbb{N}$, we define $x_n = \max \{\max_{j<n} x_j, \max_{j<n} \mathcal{G}_j(x_1, \ldots, x_j)\} + 1$ and $E_n = (x_1, \ldots, x_n)$. Lastly, we let $K = \cup_{n \in \mathbb{N}} x_n$.

First, notice that since $x_i \neq x_j, \forall i, j \in \mathbb{N}, i \neq j$, it holds that $|K| = \infty$, hence $K \in \mathcal{L}$. Moreover, notice that $E$ is a valid enumeration of $K$. Finally, notice that for every $n \in \mathbb{N}$, the algorithm either outputs a number that does not belong to $K$ or a number that has already been enumerated. To prove that formally, it suffices to show that for all $n \in \mathbb{N}$ it holds that $\mathcal{G}_n(x_1, \ldots, x_n) \notin \{x_{n+1}, x_{n+2}, \ldots\}$. This follows by two observations: for all $n \in \mathbb{N}$ it holds that $x_{n+1} > \mathcal{G}_n(x_1, \ldots, x_n)$ and $x_{n+1+j} > x_{n+1}, \forall j \in \mathbb{N}$. $\qquad\square$

It is natural to consider whether randomization can help circumvent the result of Proposition A.1. Our next result shows that this is not the case.

**Proposition A.2.** *Let $\Sigma^* = \mathbb{N}$ and $\mathcal{L}$ be the set of all infinite subsets of $\mathbb{N}$. Then, for every randomized algorithm $\mathcal{G} = (\mathcal{G}_n \colon \mathbb{N}^n \to \Delta(\mathbb{N}))_{n \in \mathbb{N}}$ there exists a language $K \in \mathcal{L}$ and an enumeration $E$ of $K$ such that, with probability 1, $\mathcal{G}$ will produce unseen elements of $K$ only finitely many times.*

*Proof.* Let $\mathcal{G} = (\mathcal{G}_n \colon \mathbb{N}^n \to \Delta(\mathbb{N}))_{n \in \mathbb{N}}$ be a (potentially randomized) generating algorithm. We will show that there is a language $K \in \mathcal{L}$ and an enumeration $E$ of $K$, both of which depend on $\mathcal{G}$, for which $\mathcal{G}$ fails to generate from $K$ in the limit, with probability 1. We will construct the enumeration and the target language inductively. For all $n \in \mathbb{N}$, we denote by $E_n$ the first $n$ elements of $E$.

Let $x_1 = 1$ and $E_1 = (1)$. For any $n \in \mathbb{N}, n \geq 2$ we define the random variable

$$X_n := \max \left\{ \max_{j<n} x_j, \max_{j<n} \mathcal{G}_j(x_1, \ldots, x_j) \right\},$$

and we let

$$x_n := \min \left\{ j \in \mathbb{N} \colon \Pr[X_n \geq j] \leq \frac{1}{n^2} \right\}.$$

We also define $E_n := (x_1, \ldots, x_n)$ and $K = \cup_{n \in \mathbb{N}} x_n$.

First, notice that, with probability 1, $x_i \neq x_j, \forall i, j \in \mathbb{N}, i \neq j$, hence it holds that $|K| = \infty$, thus $K \in \mathcal{L}$. Moreover, notice that $E$ is a valid enumeration of $K$. Next, we argue that with probability 1, the algorithm outputs unseen elements of $K$ only finitely many times. For all $n \in \mathbb{N}$, let $\mathscr{E}_n$ be the event that $\mathcal{G}_n(x_1, \ldots, x_n) \in K \setminus \{x_1, \ldots, x_n\}$. Notice that, by definition of the enumeration $E$ and the target language $K$,

$$\Pr[\mathscr{E}_n] = \Pr[\mathcal{G}_n(x_1, \ldots, x_n) \in \{x_{n+1}, x_{n+2}, \ldots\}].$$

Moreover, since with probability 1, $x_{n+1} < x_{n+2} < \ldots$, it holds that

$$\Pr[\mathcal{G}_n(x_1, \ldots, x_n) \in \{x_{n+1}, x_{n+2}, \ldots\}] \leq 1 - \Pr[\mathcal{G}_n(x_1, \ldots, x_n) < x_{n+1}].$$

By definition of $x_{n+1}$ it holds that

$$\Pr\left[\mathcal{G}_n(x_1,\ldots,x_n) < x_{n+1}\right] \geq 1 - \frac{1}{(n+1)^2}\,.$$

Chaining the previous inequalities, we get

$$\Pr\left[\mathscr{E}_n\right] \leq \frac{1}{(n+1)^2}\,,$$

thus,

$$\sum_{n\in\mathbb{N}} \Pr\left[\mathscr{E}_n\right] < \infty\,.$$

By the Borel–Cantelli lemma we deduce that with probability 1 only finitely many of the events $\{\mathscr{E}_n\}_{n\in\mathbb{N}}$ will occur.

$\square$

## A.2 Extensions to a Family of Constructions

We first explain some high-level properties of our lower bound construction and illustrate how they can be used to show non-generatability in other settings. In our construction, we can split the underlying collection $\mathcal{L}$ into two parts, say $\mathcal{L}_1, \mathcal{L}_2$ that allow for the following lower bound argument: the adversary can start enumerating some language $L_1$ from $\mathcal{L}_1$, if at some point the generator generates an unseen element of $L_1$ the adversary moves to enumerating a language $L_2$ from $\mathcal{L}_2$, and then if the generator generates unseen elements of $L_2$ the adversary moves back again to enumerating some language $L_3$ of $\mathcal{L}_1$. Crucially, the adversary can ensure that when switching from $\mathcal{L}_2$ to $\mathcal{L}_1$ the generator makes a mistake, and if there are infinitely many switches the adversary enumerates some language $L_\infty$ that is in the collection.

To illustrate the generality of these conditions, we present another family of language collections $\mathcal{L}_1, \mathcal{L}_2, \ldots, \mathcal{L}_\infty$ that are individually generatable, but whose union is not generatable. Let the domain be $\mathcal{X} = \mathbb{N} \times \mathbb{N}$. Fix an index $i \in \mathbb{N}$. We begin by defining $\mathcal{L}_i$. Each language $L \in \mathcal{L}_i$ is parameterized by $i$ finite subsets $A_{-1}, A_1, A_2, \ldots, A_{i-1}$:

$$(A_{-1}, A_1, \ldots, A_{i-1}\colon |A_{-1}|, |A_1|, \ldots, |A_{i-1}| < \infty \text{ and } A_{-1}, A_1, \ldots, A_{i-1} \subseteq \mathbb{N})\,.$$

Given the finite sets $A_{-1}, A_1, A_2, \ldots, A_{i-1} \subseteq \mathbb{N}$, the corresponding language $L$ in $\mathcal{L}_i$ is defined as follows:

$$L = \left(\bigcup_{k=1}^{i-1} \bigcup_{\ell \in A_i} \{(k,\ell)\}\right) \cup \bigcup_{j=1}^{\infty} \{(i,j)\} \cup \bigcup_{\ell \in A_{-1}} \{(-1,\ell)\}\,.$$

Next, we define $\mathcal{L}_\infty$: Each language $L \in \mathcal{L}_\infty$ is parameterized by a countable collection of finite subsets

$$(A_{-1}, A_1, A_2, \ldots\colon |A_{-1}|, |A_1|, |A_2|, \cdots < \infty \text{ and } A_{-1}, A_1, A_2, \cdots \subseteq \mathbb{N})\,.$$

Given finite sets $A_{-1}, A_1, A_2, \cdots \subseteq \mathbb{N}$, the corresponding language $L$ in $\mathcal{L}_\infty$ is defined as follows

$$L = \left(\bigcup_{k=1}^{\infty} \bigcup_{\ell \in A_i} \{(k,\ell)\}\right) \cup \bigcup_{\ell \notin A_{-1}} \{(-1,\ell)\}\,.$$

Note that the last union is over elements $\ell$ not belonging to $A_{-1}$. Hence, each language in $\mathcal{L}_\infty$, contains all but finitely many elements of the form $(-1,j)$ (for $j \in \mathbb{N}$). This, in particular, ensures that $\mathcal{L}_\infty$ can be trivially generates: e.g., it suffices to output element $(-1,t)$ in the $t$-th iteration.

*Remark* A.3. Consider the collection $\mathcal{L} = \cup_{i=1}^{\infty}\mathcal{L}_i \cup \mathcal{L}_\infty$ Having described this collection, it is not hard to show that a direct adaptation of our approach in the main result goes through. We can group these collections in a natural way, *e.g.*, by having $\mathcal{L}_\infty, \mathcal{L}_1, \mathcal{L}_3, \mathcal{L}_5, \ldots$, in one group and $\mathcal{L}_2, \mathcal{L}_4, \ldots$, in the other. Then, the lower bound follows by picking an enumeration that alternates between these two collections, as described above.

# B Proofs

## B.1 Proof of Theorem 3.1

In this section, we prove Theorem 3.1. We first state a more detailed version of the theorem.

**Theorem B.1.** *Let $\Sigma^* = \mathbb{Z}$ and $\mathcal{L} = \mathcal{L}_1 \cup \mathcal{L}_2$ where*

$$\mathcal{L}_1 := \{L_{A,B} := (\mathbb{Z}_- \setminus A) \cup (\mathbb{Z}_+ \setminus B), A \subseteq \mathbb{Z}_-, |A| < \infty, B \subseteq \mathbb{Z}_+, |B| = \infty\} \,,$$
$$\mathcal{L}_2 := \{L_{A,B} := (\mathbb{Z}_- \setminus A) \cup (\mathbb{Z}_+ \setminus B), A \subseteq \mathbb{Z}_-, |A| = \infty, B \subseteq \mathbb{Z}_+, |B| < \infty\} \,.$$

*Then, $\mathcal{L}_1, \mathcal{L}_2$ are trivially non-uniformly generatable and $\mathcal{L}$ is not generatable.*

*Proof.* Notice that every language $L_{A,B}$ in $\mathcal{L}_1$ contains all the negative integers except for the finite set $A$ and all of the positive integers except for the infinite set $B$. The languages in $\mathcal{L}_2$ are defined symmetrically. Hence, the algorithm can generate from $\mathcal{L}_2$ without observing any input samples simply by just omitting increasing prefixes of the positive integers (symmetrically for $\mathcal{L}_2$.)

Next, we show that $\mathcal{L} = \mathcal{L}_1 \cup \mathcal{L}_2$ is not generatable. Assume, towards a contradiction, that there exists a deterministic generating algorithm

$$\mathcal{G} = (\mathcal{G}_1, \mathcal{G}_2, \dots)$$

that generates from $\mathcal{L}$ in the limit. We now describe an adversarial strategy that constructs an enumeration $E = \{w_1, w_2, \dots\}$ of a target language $K \in \mathcal{L}$ such that the generator makes infinitely many mistakes.

To simplify the notation define the following sets:

- $S_t^+$ and $S_t^-$ denote the sets of positive and negative numbers enumerated up to time $t$, respectively, *i.e.*, $S_t^+ = \{w_1, \dots, w_t\} \cap \mathbb{Z}_+, S_t^- = \{w_1, \dots, w_t\} \cap \mathbb{Z}_-$;

- $P_t$ be the set of positive integers outputted by the generator in rounds 1 through $t$.

The adversary's construction is organized into *phases*, each divided into two subphases.

**Phase 1.** This phase is divided into two sub-phases.

1. **Subphase 1-A:** In every round $t$, the adversary enumerates the positive integer $t$, *i.e.*,

$$w_t = t.$$

   This subphase continues as long as the generator's output, given the current prefix $(1, 2, \dots, t)$, is *not* an integer greater than $t$. If there is a first round $t$ for which

$$\mathcal{G}_t(1, 2, \dots, t) \in \mathbb{Z}_+ \setminus \{1, \dots, t\} \,,$$

   then subphase 1-A ends, and we move to Subphase 1-B. If the generator never deviates (*i.e.*, never outputs an integer greater that $\{1, \dots, t\}$), then the enumeration $E = \{1, 2, 3, \dots\}$ is complete for the language $K = \mathbb{N}$, and $\mathbb{N} \in \mathcal{L}_2$. Moreover, by definition, the generator makes infinitely many mistakes (because it never produces an unseen element of $K$).

2. **Subphase 1-B:** Let $t_1$ denote the first time step at which we enter this subphase. In every round $t$ that we are in this phase, the adversary enumerates the negative integer

$$w_t = -t - t_1 - 1 \,.$$

   This subphase continues until the first time $t$ at which the generator outputs a negative number that is *smaller* than the current $w_t$, *i.e.*,

$$\mathcal{G}_t(S_t) \in \mathbb{Z}_- \quad \text{and} \quad \mathcal{G}_t(S_t) < w_t \,.$$

   If no such round occurs, then the final enumeration $E$ will be complete for some $K \in \mathcal{L}_1$, since it consists of all negative integers and a finite number of positive integers, and the generator will have made infinitely many mistakes.

**Phase $k$ ($k \geq 2$).** After the end of Subphase $(k-1)$-B, the adversary alternates the strategy as follows:

1. **Subphase $k$-A:** Let $t_k$ be the first time of Subphase $k$-A. The adversary first enumerates a fresh positive number defined by

$$p_k = 1 + \max\{x : x \in S_{t_k}^+ \cup P_{t_k}\}.$$

Then, in every round $t = t_k, t_k + 1, \ldots$ during this subphase, the adversary enumerates

$$w_t = p_k + (t - t_k),$$

thereby listing an increasing sequence of fresh positive numbers. This subphase ends when, for the first time, the generator outputs a positive number (given the current prefix) that exceeds the maximum of the current enumerated positives. An identical argument to the one we used for subphase 1-A shows that if this subphase never terminates, then the generator makes infinitely many mistakes and the adversary gives a complete enumeration of a language from $\mathcal{L}_2$.

2. **Subphase $k$-B:** After completing Subphase $k$-A, the adversary enters Subphase $k$-B. In every round during this subphase, the adversary enumerates the number

$$w_t = \min S_t^- - 1,$$

introducing a fresh negative number. This subphase ends at the first round when the generator outputs a negative number that is less than $w_t$. An identical argument to the one we used for subphase 1-B shows that if this subphase never terminates, then the generator makes infinitely many mistakes and the adversary gives a complete enumeration of a language from $\mathcal{L}_1$.

Let us now assume that infinitely many of the phases are executed. Then, the target language is $K = \mathbb{Z}_- \cup A$, where $A \subset \mathbb{N}$, is determined by the elements enumerated during the subphases $k$-A, $k \in \mathbb{N}$, hence it is a valid language from $\mathcal{L}_1$. Moreover, notice that every time we transition from subphase $k$-B to $(k+1)$-A, then the generator makes a mistake because the element it outputted is not included in the constructed enumeration.

Since there are infinitely many phases, and in each phase the generator is forced to err at least once, the deterministic generating algorithm makes infinitely many mistakes. This contradicts the definition of generation in the limit. $\qquad\square$

*Remark* B.2 (Language Identification Is Not Closed under Finite Unions Either). It is worth highlighting that identification in the limit is not closed under finite unions either and it exhibits a very similar behavior: there is a finite collection $\mathcal{L}_1$ that is trivially uniformly identifiable and a countable collection $\mathcal{L}_2$ that is trivially non-uniformly identifiable such that $\mathcal{L}_1 \cup \mathcal{L}_2$ is not identifiable in the limit. To see that, let $\mathcal{L}_1 = \{\mathbb{N}\}$ and $\mathcal{L}_2 = \{l_i := \{1, \ldots, i\}, i \in \mathbb{N}\}$.

## B.2 Proof of Theorem 3.2

We first restate a more detailed version of the theorem for completeness.

**Theorem B.3.** *Let $\Sigma^* = \mathbb{Z}$ and let $\mathcal{L} = \mathcal{L}_1 \cup \mathcal{L}_2$ where*

$$\mathcal{L}_1 = \{L_A := \mathbb{Z}_- \cup A : A \subseteq \mathbb{Z}_+\}$$
$$\mathcal{L}_2 = \{\mathbb{N}, L_{i,B} := \{-i, \ldots, -1\} \cup (\mathbb{Z}_+ \setminus B) : i \in \mathbb{N}, B \subseteq \mathbb{Z}_+, |B| < \infty\}.$$

*Then,*

- *$\mathcal{L}_1$ is uncountable and trivially uniformly generatable,*

- *$\mathcal{L}_2$ is countable and trivially non-uniformly generatable, and*

- *no deterministic generating algorithm $\mathcal{G} = (\mathcal{G}_1, \mathcal{G}_2, \ldots)$ can generate from $\mathcal{L}$ in the limit.*

*Proof.* Notice that every language $L_A$ in $\mathcal{L}_1$ contains all the negative integers and some subset $A$ of the positive integers. Hence, $\cap_{L \in \mathcal{L}_1} = \mathbb{Z}_-$, which implies that an algorithm can generate from $\mathcal{L}_1$ without using any input samples, by simply outputting negative integers. Regarding $\mathcal{L}_2$, notice that every language $L_{i,B}$ contains the first $i$ negative integers and all positive integers except for the finite set $B$. We claim that for any such language $L_{i,B}$, the algorithm that just outputs number

$\{1, 2, \dots\}$ achieves generation in the limit, without requiring any input samples from $L_{i,B}$. To see that, notice that since $B$ is finite it has some maximum element, denoted by $x_B$. Notice that $(\mathbb{Z}_+ \setminus \{1, \dots, x_B\}) \subseteq L_{i,B}$. Thus, at timestep $t = x_B + 1$ the algorithm will generate a valid element and will keep generating valid elements from that point on. Hence, $\mathcal{L}_2$ is trivially non-uniformly generatable. He next show that $\mathcal{L} = \mathcal{L}_1 \cup \mathcal{L}_2$ is not generatable. The proof of this uses an analog of the diagonalization argument in Theorem 3.1. We produce the complete argument below.

Assume, towards a contradiction, that there exists a deterministic generating algorithm

$$\mathcal{G} = (\mathcal{G}_1, \mathcal{G}_2, \dots)$$

that generates from $\mathcal{L}$ in the limit. We now describe an adversarial strategy that constructs an enumeration $E = \{w_1, w_2, \dots\}$ of a target language $K \in \mathcal{L}$ such that the generator makes infinitely many mistakes.

For clarity, let:

- $S_t^+$ and $S_t^-$ denote the sets of positive and negative numbers enumerated up to time $t$, respectively, *i.e.*, $S_t^+ = \{w_1, \dots, w_t\} \cap \mathbb{Z}_+$, $S_t^- = \{w_1, \dots, w_t\} \cap \mathbb{Z}_-$;

- $P_t$ be the set of positive integers outputted by the generator in rounds 1 through $t$.

The adversary's construction is organized into *phases*, each divided into two subphases.

**Phase 1.**

1. **Subphase 1-A:** In every round $t$, the adversary enumerates the positive integer $t$, *i.e.*,

$$w_t = t.$$

This subphase continues as long as the generator's output, given the current prefix $(1, 2, \dots, t)$, is *not* an integer greater than $t$. If there is a first round $t$ for which

$$\mathcal{G}_t(1, 2, \dots, t) \in \mathbb{Z}_+ \setminus \{1, \dots, t\},$$

then subphase 1-A ends, and we move to Subphase 1-B. If the generator never deviates (*i.e.*, never outputs an integer greater that $\{1, \dots, t\}$), then the enumeration $E = \{1, 2, 3, \dots\}$ is complete for the language $K = \mathbb{N}$, and $\mathbb{N} \in \mathcal{L}_2$. Moreover, by definition, the generator makes infinitely many mistakes (because it never produces an unseen element of $K$).

2. **Subphase 1-B:** Let $t_1$ denote the first time step at which we enter this subphase. In every round $t$ that we are in this phase, the adversary enumerates the negative integer

$$w_t = -t - t_1 - 1.$$

This subphase continues until the first time $t$ at which the generator outputs a negative number that is *smaller* than the current $w_t$, *i.e.*,

$$\mathcal{G}_t(S_t) \in \mathbb{Z}_- \quad \text{and} \quad \mathcal{G}_t(S_t) < w_t.$$

If no such round occurs, then the final enumeration $E$ will be complete for some $K \in \mathcal{L}_1$, since it consists of all negative integers and a finite number of positive integers, and the generator will have made infinitely many mistakes.

**Phase $k$ ($k \geq 2$).** After the end of Subphase $(k-1)$-B, the adversary alternates the strategy as follows:

1. **Subphase $k$-A:**

- Let $t_k$ be the first time of Subphase $k$-A. The adversary first enumerates a fresh positive number defined by

$$p_k = 1 + \max\{x : x \in S_{t_k}^+ \cup P_{t_k}\}.$$

- Then, in every round $t = t_k, t_k + 1, \dots$ during this subphase, the adversary enumerates

$$w_t = p_k + (t - t_k),$$

thereby listing an increasing sequence of fresh positive numbers.

- This subphase ends when, for the first time, the generator outputs a positive number (given the current prefix) that exceeds the maximum of the current enumerated positives. An identical argument to the one we used for subphase 1-A shows that if this subphase never terminates, then the generator makes infinitely many mistakes and the adversary gives a complete enumeration of a language from $\mathcal{L}_2$.

2. **Subphase $k$-B:**

    - After completing Subphase $k$-A, the adversary enters Subphase $k$-B. In every round during this subphase, the adversary enumerates the number

    $$w_t = \min S_t^- - 1,$$

    introducing a fresh negative number.
    - This subphase ends at the first round when the generator outputs a negative number that is less than $w_t$. An identical argument to the one we used for subphase 1-B shows that if this subphase never terminates, then the generator makes infinitely many mistakes and the adversary gives a complete enumeration of a language from $\mathcal{L}_1$.

Let us now assume that infinitely many of the phases are executed. Then, the target language is $K = \mathbb{Z}_- \cup A$, where $A \subset \mathbb{N}$, is determined by the elements enumerated during the subphases $k$-A, $k \in \mathbb{N}$, hence it is a valid language. Moreover, notice that every time we transition from subphase $k$-B to $(k+1)$-A, then the generator makes a mistake because the element it outputted is not included in the constructed enumeration.

Since there are infinitely many phases, and in each phase the generator is forced to err at least once, the deterministic generating algorithm makes infinitely many mistakes. This contradicts the definition of generation in the limit. $\qquad\square$

## B.3    Proof of Theorem 3.3

In this section, we prove Theorem 3.3. We begin by stating a more detailed version of Theorem 3.3.

**Theorem B.4.** *Let $\Sigma^* = \mathbb{Z}$ and let*

$$\mathcal{L} := \{L_{A,B} := (\mathbb{Z}_- \setminus A) \cup B, A \subseteq \mathbb{Z}_-, |A| < \infty, B \subseteq \mathbb{Z}_+\} .$$

*Then, $\mathcal{L}$ is is uncountable and trivially non-uniformly generatable but it does not satisfy the EUC property.*

*Proof.* For each one of uncountably many subsets $B \subseteq \mathbb{Z}_+$, $\mathcal{L}$ contains a language (e.g., $\mathbb{Z}_- \cup B$) and, hence, $\mathcal{L}$ is an uncountable collection. To see that $\mathcal{L}$ is trivially non-uniformly generatable notice that, since every $K \in \mathcal{L}$ omits only finitely many negative integers the algorithm that in every round $t$ generates the largest element from the set $O_t = \{i \in \mathbb{Z}_-, i \leq -t, i \notin S_t\}$ generates in the limit. In fact, this algorithm generates in the limit in the modified setting where $S_t = \emptyset$ for all rounds $t$, hence it is trivially uniformly generatable.

We now show that $\mathcal{L}$ does not satisfy the EUC property. Consider any target language $K$ and $S_t$ any set of elements of $K$. Consider the induced version space $V(\mathcal{L}, S_t)$, *i.e.*, the set of languages from $\mathcal{L}$ containing $S_t$. Assume that $\cap_{L \in V(\mathcal{L}, S_t)} L \neq S_t$ for some $t$. Then, there exists some $x_t \in \cap_{L \in V(\mathcal{L}, S_t)} L$ such that $x_t \notin S_t$. Consider the language $L = \mathbb{Z}_- \setminus \{x_t\}$. Then, $S_t \subseteq L$ and $x_t \notin L$, hence this show that $\cap_{L \in V(\mathcal{L}, S_t)} L = S_t$ (violating the EUC property). $\qquad\square$

