# OpenReview forum: "On Union-Closedness of Language Generation"
_NeurIPS.cc/2025/Conference — NeurIPS 2025 poster_

### Official Review · Reviewer_B1o9 · 2025-06-25

**Clarity:** 3
**Significance:** 4
**Originality:** 4
**Rating:** 5
**Confidence:** 1

**Summary:**

The paper studies language generation in the limit (LGitL), the on-line model introduced by Kleinberg & Mullainathan (2024) and refined by Li et al. (2024).

Where previous work showed that every countable collection of languages is generatable, the authors focus on finite unions of (possibly uncountable) collections and prove three main theorems:

1. Non-closure under finite unions (Thm 3.1). They construct two uncountable, non-uniformly-generatable collections L1,L2L1​,L2​ whose union is not generatable.

2. A "minimal" counter-example (Thm 3.2). They sharpen the above by making L1L1​ countable and L2L2​ uniformly generatable (and generatable without seeing any data) while their union remains non-generatable.

3. Non-uniform generation without EUC (Thm 3.3). They give an uncountable class that is non-uniformly generatable yet violates the Eventually-Unbounded-Closure property, answering an open question of Li et al.

Technically, the work introduces a two-sided, phased diagonalisation argument that alternates mistakes across even/odd rounds, carefully ensuring that each collection on its own is easy, yet their union defeats any generator. The construction relies on highly artificial languages such as "(almost) all negative integers union an infinite co-finite subset of the positives."

**Questions:**

I'm not really an expert in this space, so my questions aren't great.

Primarily I'm interested in if there is going to be more discussion of the claim that ensembling LLMs doesn't work.

**Ethical Concerns:**

["NO or VERY MINOR ethics concerns only"]

**Final Justification:**

I raised my scores in response to the thoughtful feedback from the author. I acknowledge my relative ignorance in this space compared to others (low confidence)

**Limitations:**

Yes

**Quality:**

4

**Strengths And Weaknesses:**

Strengths

This paper tackles a natural and previously open question in the theory of language generation in the limit: whether the class of generatable collections is closed under finite union. The authors answer this in the negative and, in doing so, settle three distinct open problems posed by Li et al. (2024). The work is technically original! the 'phase-split' diagonalisation that alternates mistakes across odd and even rounds is genuinely clever and, to my knowledge, new to the literature. That technique not only establishes the main non-closure result but is also likely to be useful in future lower-bound constructions for language-generation theory. The paper demonstrates solid depth too... the constructions simultaneously keep each collection individually easy to generate from while making their union provably impossible. The authors situate their results well within the recent surge of work on LGitL, giving readers a clear map of what was known and what remained mysterious.

Weaknesses

The paper operates in an extremely stringent model that demands eventual perfect generation. After some unknown time every output must belong to the target language. Real-world language-model ensembles are inherently probabilistic and tolerate small error rates, so it remains unclear whether the impossibility result persists in a more forgiving, approximate-generation setting. A related concern is the artificial nature of the language families used in the proofs. They are built from sets such as 'all non-positive integers plus an infinite co-finite subset of the positives', which bear little resemblance to natural language or even to the customary formal languages of automata theory. It is therefore an open question whether the same pathology could occur for structured families like regular or context-free languages. In addition, while the authors motivate their study with the prospect of combining large language models, the perfect-generation assumption severs most practical connections, and the paper does not examine approximate or probabilistic ensemble strategies that practitioners actually use. Finally, presentation could be smoother: notation for phase indices is heavy, several typos remain, and some late-section forward references disrupt the reading flow. Addressing these points would significantly strengthen both the accessibility and the practical relevance of the contribution."

---

> ### Author Rebuttal · Authors · 2025-07-30
>
> Thank you for your careful review. We are happy to see that you found the techniques clever and likely to be useful in the future. Please find our responses to your comments below.
>
> > Stringent model that demands eventual perfect generation.
>
> While we agree that the model might initially seem stringent, there have been several surprising positive results in this model: [KM’24] showed that all countable collections are generatable, [RR’25] extended this to the “noisy” setting, and [PRR’25, KW’25] demonstrated that generation “with breadth” can be achieved in this model. Given these positive results, one might expect that finite unions of generatable classes are generatable, but our result shows that this is not the case. We believe this reveals a fundamental limitation that is relevant to the NeurIPS community, especially given the recent positive results on language generation published at NeurIPS and ICML.
>
> [KM’24] Jon Kleinberg and Sendhil Mullainathan. Language generation in the limit. In Advances in Neural Information Processing Systems, Volume 37, 2024
>
> [RR’25] Ananth Raman and Vinod Raman. Generation from noisy examples. In Forty-second International Conference on Machine Learning, 2025
>
> [PRR’25] Charlotte Peale, Vinod Raman, and Omer Reingold. Representative language generation. In Forty-second International Conference on Machine Learning, 2025
>
> [KW’25] Jon Kleinberg and Fan Wei. Density Measures for Language Generation. In IEEE 65th Annual Symposium on Foundations of Computer Science (FOCS).
>
> > Whether the same pathology could occur for structured families like regular or context-free languages.
>
>  Exploring whether non-union closeness could occur for more structured families and understanding which properties of the language collections lead to closeness under unions is an interesting open question left by our work. That said, the complicated nature of the families in our lower bound is perhaps expected since the model of language generation gives generators substantial power: there are no computational restrictions and no bound on the amount of training data used by them. It is possible that once one places computational restrictions and bounds on the training data, then non-union closeness could be witnessed by simple families; and this is also an interesting open question.
>
> > Perfect-generation assumption.
>
> Please see our response to the first question above.
>
> > Presentation … addressing these points would significantly strengthen … accessibility.
>
> Thank you for the suggestions, we will fix the typos, referencing errors, and simplify notation in the final version.
>
>
> > In addition, while the authors motivate their study with the prospect of combining large language models, the perfect-generation assumption severs most practical connections, and the paper does not examine approximate or probabilistic ensemble strategies that practitioners actually use
>
> A strength of the Kleinberg–Mullainathan model is that it places no assumptions on the generators – neither architectural nor computational. At this level of generality, theoretical results inevitably require assumptions like perfect-generation.
> Nevertheless, we believe our core insight that "merging" LLMs is fundamentally difficult could remain valid in practice: here, our result shows that unlike classification (where methods like XGBoost and Random Forests can divide tasks into simpler sub-problems), LLM training cannot always be decomposed this way. For instance, one might hope to train separate models on $\mathcal{L_1}$ and $\mathcal{L_2}$, then merge them to handle $\mathcal{L_1} \cup \mathcal{L_2}$ – but our result shows this approach is not feasible in general.
>
> This, perhaps, offers a principled explanation for challenges in “merging” LLMs in practice. For instance, a recent survey [1] notes that "guaranteeing the performance of model merging remains challenging" and requires models to be "fine-tuned based on the same pre-trained model, with careful control over parameters." Even then, "there is still a significant gap between the merged and independent models.
>
> [1] Model Merging in LLMs, MLLMs, and Beyond: Methods, Theories, Applications and Opportunities. Enneng Yang, Li Shen, Guibing Guo, Xingwei Wang, Xiaochun Cao, Jie Zhang, Dacheng Tao.

---

### Official Review · Reviewer_HnRj · 2025-06-30

**Clarity:** 4
**Significance:** 4
**Originality:** 4
**Rating:** 6
**Confidence:** 3

**Summary:**

This paper shows that generatable language classes are not closed under union, and resolves other open questions about the nature of language generation in the limit.

**Questions:**

Typo on Line 82 (extra period inserted after "language")

Equations after 366 defining $L_1'$ and $L_2'$
should have $A \subseteq Z_-$ and $B \subseteq Z_+$ (signs are flipped).

The authors posit that the non-closure under union has consequences for the generation equivalent of boosting. This seems like an important finding, but it is not yet clear to me what "combining generators" looks like. Does this apply to *learning* generators, e.g., training LLMs? Or does it have implications for things like ensembling pretrained LLMs at inference time? I'd love to see more discussion here.

I would also like to hear more about how this results shows how hard *characterizing* generatibility is... does this mean that generatability might not be a very useful framework for understanding language learning because it has difficult properties? Or is it still useful despite its challenging properties?

**Ethical Concerns:**

["NO or VERY MINOR ethics concerns only"]

**Final Justification:**

My questions were adequately addressed (specifically, questions about the implications of the findings) and I am happy to recommend this paper for publication.

**Limitations:**

The authors might include a section clarifying their views on how *practically* useful their results are. Overall, I do not see negative societal impact from this kind of theoretical work.

**Quality:**

4

**Strengths And Weaknesses:**

The paper is surprisingly easy to follow, and is well organized pedagogically. I enjoyed reading it.

In terms of significance, this paper is exciting because it establishes an important difference between statistical learning and language generation. This has important theoretical implications, though it is unclear what the practical implications will be.

The proof sketch provided in the main text appears to be correct, as far as I can tell.

---

> ### Author Rebuttal · Authors · 2025-07-30
>
> Thank you for the strong support of our paper. We are happy to see that you found the paper easy to read and significant. We respond to your specific comments and questions below and will include this information in the final version.
>
> > Typos.
>
> Thank you for reading the paper carefully, we will fix all of them.
>
> > Does this apply to learning generators, e.g., training LLMs? Or does it have implications for things like ensembling pretrained LLMs at inference time?
>
> Our view is that our result's main implication is for training LLMs: it shows that unlike classification, LLM training cannot always be divided into simpler sub-tasks where we train separate models and then combine them (as is possible with XGBoost and Random Forests in classification). For instance, to train a model for $\mathcal{L_1} \cup \mathcal{L_2}$, one might train two models in parallel – one for $\mathcal{L_1}$ and one for $\mathcal{L_2}$ – then merge them.  Our result shows this approach isn't feasible in general.
>
> Perhaps the most closely related practical instance involves challenges in "merging" LLMs. A recent survey [1] notes:
> "In practical settings, guaranteeing the performance of model merging remains challenging" and requires "[all models to be] fine-tuned based on the same pre-trained model, with careful control over [parameters]." Moreover, "[even after careful parameter selection] there is still a significant gap between the merged and independent models."
> Our results offer a principled explanation for these challenges.
>
> [1] Model Merging in LLMs, MLLMs, and Beyond: Methods, Theories, Applications and Opportunities. Enneng Yang, Li Shen, Guibing Guo, Xingwei Wang, Xiaochun Cao, Jie Zhang, Dacheng Tao.
>
> > How this result shows how hard characterizing generatibility is.
>
> Our results show that any characterization of generatability must ensure that generatability is not closed under finite unions. This prohibits one from using most of the existing characterizations/dimensions in statistical learning theory (e.g., the VC dimension, the Littlestone dimension, the recent DS dimension, and many others) or their variations because all of them ensure closure under finite unions.
>
> > Does this mean that generatability might not be…useful…because it has difficult properties? Or is it still useful…?
>
> We believe generatability is a useful notion despite having difficult properties (e.g., not being closed under finite-unions) because:
> - It arises from a simple and (arguably) natural definition (given a sequence of strings from a language, can the generator learn to generate valid and unseen strings?), and
> - It turns out to have different properties than standard prediction tasks, which suggests that studying generatability could lead to new theoretical techniques that can have other applications.
>
> > The authors might include a section clarifying their views on how practically useful their results are.
>
> Thanks for the suggestion, we will use the additional page provided in the final version to include this section; please see our response to ``Does this apply … inference time?’’ above.

---

> > ### Comment · Reviewer_HnRj · 2025-08-05
> >
> > Thank you for your thoughtful responses. I look forward to reading the additional section with discussion of the implications of your findings.

---

### Official Review · Reviewer_dkh6 · 2025-07-01

**Clarity:** 3
**Significance:** 3
**Originality:** 3
**Rating:** 5
**Confidence:** 2

**Summary:**

This paper investigates the theoretical foundations of language generation in the limit, a setting introduced by Kleinberg and Mullainathan (2024) and extended by Li et al. (2024). The authors focus on the union-closedness properties of generatable language classes, addressing three open questions posed by Li et al. Specifically, they show that the class of generatable (and even non-uniformly generatable) languages is not closed under finite unions, disproving the possibility of combining generators to construct a more powerful one. They further construct a minimal example where the union of a uniformly and a non-uniformly generatable class is not generatable, despite each being generatable individually. Finally, they provide a concrete uncountable class that is non-uniformly generatable but violates the Eventually Unbounded Closure (EUC) condition. This work novelty lies in a new diagonalization technique that ensures individual generatability while rendering the union non-generatable.

**Questions:**

Clarification on Practical Implications:
While the theoretical contribution is strong, could the authors comment on whether similar non-union-closedness phenomena might manifest in practical language generation systems (e.g., autoregressive LLMs)? Any intuition or empirical proxy would help ground the theory.

Visualization or Simplified Example:
Even a toy example illustrating how a generator fails under the constructed union could greatly aid intuition. Would the authors consider including such a figure or pseudocode in the final version?

**Ethical Concerns:**

["NO or VERY MINOR ethics concerns only"]

**Final Justification:**

Thank you for the clear and constructive rebuttal. The added discussion on how finite-union non-closedness relates to practical LLM merging challenges satisfactorily addresses my earlier question on real-world relevance.

I particularly welcome the plan to include an intuitive “majority-vote generator” example with figures and pseudocode. A minimal, self-contained demonstration will make the core idea far more accessible to readers outside the immediate subfield, and will strengthen the paper’s pedagogical value.

Given the strength of the theoretical contribution, the resolution of multiple open questions, and the authors’ commitment to improving clarity with concrete examples, I am raising my score from 4 (borderline accept) to 5 (accept).

**Limitations:**

yes

**Quality:**

3

**Strengths And Weaknesses:**

Strengths
- The paper resolves three open questions in the theory of language generation in the limit, demonstrating both depth and relevance.
- The construction of counter examples is technically elegant and novel, especially the diagonalization argument.
- The results reveal fundamental differences between generation and traditional learning frameworks (e.g., lack of union-closedness), which is conceptually significant.

Weaknesses
- No empirical or illustrative examples are provided to help visualize the main ideas.

---

> ### Author Rebuttal · Authors · 2025-07-30
>
> Thank you for supporting our paper. We are very happy to see that you appreciated the depth and elegance of our techniques. We answer your specific questions below and hope you will strengthen your support for the paper.
>
> > While the theoretical contribution is strong…practical language generation systems (e.g., autoregressive LLMs)?
>
>  That is an excellent question. Since the Kleinberg–Mullainathan model places no restrictions on generators, our impossibility result applies to autoregressive LLMs and other language generation systems. While practical data is non-adversarial (so our results don't directly apply), LLM merging has proven challenging in practice. A recent survey [1] notes that "guaranteeing the performance of model merging remains challenging" and requires "[all models to be] fine-tuned based on the same pre-trained model, with careful control over parameters." Even then, "there is still a significant gap between the merged and independent models." Our results provide principled grounds for why merging LLMs is much more challenging than merging classifiers, where methods like XGBoost and random forests succeed and are supported by theoretical results showing classification is closed under finite unions. We will include these points in the final version.
>
> [1] Model Merging in LLMs, MLLMs, and Beyond: Methods, Theories, Applications and Opportunities. Enneng Yang, Li Shen, Guibing Guo, Xingwei Wang, Xiaochun Cao, Jie Zhang, Dacheng Tao.
>
> > Would the authors consider including such a figure or pseudocode in the final version?
>
> Thank you for the suggestion. We will add examples with figures and pseudocode illustrating how natural generators fail on the constructed union. Here's one example:
>
> *Majority-Vote Based Generator.* A natural strategy for generating from $\mathcal{L}_1\cup \mathcal{L}_2$ is combining generators for $\mathcal{L}_1$ and $\mathcal{L}_2$, similar to how boosting combines weak learners in classification.
> This can be done in several ways. For instance, one approach is to run the generators independently and define a “scoring” function which is used as a proxy for which generator is “most successful” so far and using the output of the “most successful” generator. Since one generator makes infinitely many mistakes and the other only finitely many, one would hope that there exists a natural scoring function that will detect the one that makes finitely many mistakes. A concrete example of such a function is to count the number of (provably) correct generations a generator has made so far. Indeed, whenever a generator outputs a valid string, this string will appear at a finite time in the input stream, so we can compute such a function from data. Unfortunately, this scoring function and its natural variants failed. In fact, that result shows that no such scoring function could exist. We will include details on this (along with figures and pseudocode) in the final version.

---

### Decision · Program_Chairs · 2025-09-17

**Decision:**

Accept (poster)

**Comment:**

This is a theory paper that proves that in the Kleinberg-Mullainathan model of language generation in the limit, generatable classes are not closed under finite union, resolving three open questions. The technical core, a two‑sided phased diagonalization that forces mistakes on alternating rounds while keeping each side individually easy, seems potentially useful beyond this work. Reviewers converged positive but with low confidences, but strong substantive praise. The authors' rebuttal addressed clarity and implications and committed to adding an intuitive toy example and tightening presentation. The main drawback is external relevance: the model is exacting, the languages are artificial.